# Spatial variation and antecedent sea surface temperature conditions influence Hawaiian intertidal community structure

Rebecca J. Ward[1,2]*, T. Erin Cox[3], Anuschka Faucci[4], Florybeth Flores La Valle[5], Joanna Philippoff[6], Jessica L. B. Schaefer[7], Ian M. Ware[8], Matthew L. Knope[1,2]

1 Department of Biology, University of Hawai'i at Hilo, Hilo, HI, United States of America, 2 Tropical Conservation Biology and Environmental Science Graduate Program, University of Hawai'i at Hilo, Hilo, HI, United States of America, 3 Department of Biological Sciences, University of New Orleans, New Orleans, LA, United States of America, 4 Math & Sciences Division, Leeward Community College, Pearl City, HI, United States of America, 5 Division of Natural Sciences, Pepperdine University, Malibu, CA, United States of America, 6 Curriculum Research & Development Group, University of Hawai'i at Mānoa, Honolulu, HI, United States of America, 7 Animal Behavior Graduate Group, University of California Davis, 227 Life Sciences, Davis, CA, United States of America, 8 Institute of Pacific Islands Forestry, Pacific Southwest Research Station, USDA Forest Service, Hilo, HI, United States of America

* rebeccajay.ward@gmail.com

**Data Availability Statement:** All relevant data are within the paper: https://doi.org/10.5061/dryad.zs7h44jd8.

## Abstract

Global sea surface temperatures (SSTs) are increasing, and in Hawai'i, rates of ocean warming are projected to double by the end of the 21st century. However, current nearshore warming trends and their possible impacts on intertidal communities are not well understood. This study represents the first investigation into the possible effects of rising SST on intertidal algal and invertebrate communities across the Main Hawaiian Islands (MHI). By utilizing citizen-science data coupled with high-resolution, daily SST satellite measurements from 12 intertidal sites across the MHI from 2004–2019, the response of intertidal algal and invertebrate abundance and community diversity to changes in SST was investigated across multiple spatial scales. Results show high rates of SST warming ($0.40°C$ Decade$^{-1}$) over this study's timeframe, similar to predicted rates of warming for Hawai'i by the end of the 21st century. Changes in abundance and diversity in response to SST were variable among intertidal sites, but differences in antecedent SST among intertidal sites were significantly associated with community dissimilarity. In addition, a statistically significant positive relationship was found between SST and Simpson's diversity index, and a significant relationship was also found between SST and the abundance of six dominant taxa. For five of these six dominant taxa, antecedent SSTs over the 6–12 months preceding sampling were the most influential for describing changes to abundance. The increase in community diversity in response to higher SSTs was best explained by temperatures in the 10 months preceding sampling, and the resultant decreased abundance of dominant turf algae. These results highlight rapidly warming nearshore SSTs in Hawai'i and the longer-term effects of antecedent SSTs as significant drivers of change within Hawaiian intertidal communities. Therefore, we suggest that future research and management should consider the possibility of lagging effects of antecedent SST on intertidal communities in Hawai'i and elsewhere.

**Funding:** The author(s) received no specific funding for this work.

**Competing interests:** The authors have declared that no competing interests exist.

## Introduction

The warming of Earth's oceans has already resulted in adverse and complex consequences for marine ecosystems and is considered one of the greatest environmental crises of the 21st century [1–3]. Under the influence of anthropogenic climate change, global sea surface temperatures (SSTs) over the past century have increased dramatically [3,4]. Within the Main Hawaiian Islands (MHI), warming trends are expected to continue to increase from the historical rate of warming (1976–2005) of 0.16°C Decade$^{-1}$ to 0.39°C Decade$^{-1}$ by the end of the century (2070–2100) [2]. Warming SSTs can have widespread biological consequences that can either advantage or disadvantage individual taxa through a variety of mechanisms. For example, warming SSTs have been shown to affect: the metabolism of marine organisms [5], larval dispersal and settlement success [6], and intra- and inter-specific interactions within marine communities (e.g., sex ratios, predation rates) [7–9]. Due to its influence on the ecological and physiological processes of marine organisms, ocean temperatures can serve as a primary driver of ecological change and warrant close monitoring and further study in all marine ecosystems [10–12].

The intertidal zone sits as the ecotone between marine and terrestrial environments and is vulnerable to the influences of SST change. Many foundational concepts in ecology have been developed in the intertidal environment [13,14], which is also prized for its economic benefits [15–18], cultural significance [19], and ease of study [20,21]. Intertidal ecological communities can vary significantly across small spatial scales [22–24] and in response to the cumulative effects of ecological determinants through time. In addition to nutrient influx [25,26], substrate type [23,27], grazing and predation [22,25], and anthropogenic disturbance [28], there is compelling evidence from around the world that intertidal communities are responding to warming temperatures across multiple temporal scales. For example, short-term trends in SST (i.e., annual seasonality [29], water temperatures the few weeks prior [30–32]), have been identified as significant drivers for shifts in algal biomass, but long-term (i.e., multi-year to decadal) trends in SST warming have also been found to strongly influence intertidal communities, shifting both the range and relative abundance of populations in as little as a decade [10,33–35]. With the importance of both short and long-term SST trends, there is a need to account for multiple temporal scales when investigating the effect of SST within these systems.

Temperate continental intertidal systems (particularly in North America and Europe) have had a long history of intensive Western scientific study [13,14]. In comparison, intertidal systems in more geographically isolated locations such as the MHI have received relatively little attention until recently [e.g., 23,24,36,37]. Although Hawai'i's tidal range is limited (~1 m between the highest high and lowest low tides) [36] the intertidal zone has a diverse assemblage of fishes, algae, and invertebrates [23,37]. A few notable studies conducted within Hawaiian intertidal zones have investigated both spatial and temporal variation in community structure [23,24,37]. These studies have found high variability in Hawaiian intertidal communities both between sites and across islands [23,24], with site-level differences in substrate type [23], topography [27], and wave/wind exposure [24,38] suggested as factors influencing local community composition. One study at 'Ewa Beach on the southwest side of O'ahu identified maximum water temperature as a significant variable for explaining yearly variation in intertidal algal community composition [27]. Despite the study's limited spatial scale, it serves as a critical first step into elucidating the relationships between changing SST and Hawaiian intertidal communities [27].

If SST in Hawai'i increases according to projected trends [3], there is a need to understand how both the individual taxa and the intertidal communities will respond. Due to the large variability in physical characteristics and community composition between Hawaiian intertidal

sites [23,24], analyses limited to a single intertidal site may not be representative of the entire MHI. To effectively capture responses to SST change across the MHI, investigations conducted across multiple spatial and temporal scales with the highest resolution SST data possible are required. Furthermore, as previous studies outside of Hawai'i have shown intertidal communities to display both immediate and delayed responses to short- [31,32] and long-term [10,39] shifts in water temperature, analyses should ideally be conducted to identify time-windows when changes in SST are the most influential. This study is the first, to our knowledge, that tests for an association between short- and long-term changes in SST and intertidal community composition in Hawai'i and to test for significantly influential antecedent time windows. Specifically, we address the following three research questions: 1) Has SST changed, both for the MHI as a whole and across different intertidal sites over the course of this study timeframe (2004–2019)? 2) At the site-level, is the abundance of algal and invertebrate taxa and the diversity of these intertidal ecological communities responding to changes in SST? and 3) Across all sites combined, are there responses to changes in SST across the MHI for the abundance of algal or invertebrate taxa and/or for the diversity of these intertidal ecological communities? These questions are addressed by utilizing daily SST measurements at a 1/20° (~6 km$^2$) horizontal resolution and intertidal ecological community datasets from 12 intertidal sites across the MHI from 2005 through 2019.

## Materials and methods

### Site descriptions

Twelve intertidal sites located on the islands of O'ahu, Moloka'i, Maui, and Hawai'i (Fig 1) were selected for this study. Since 2005, benthic algal and invertebrate community

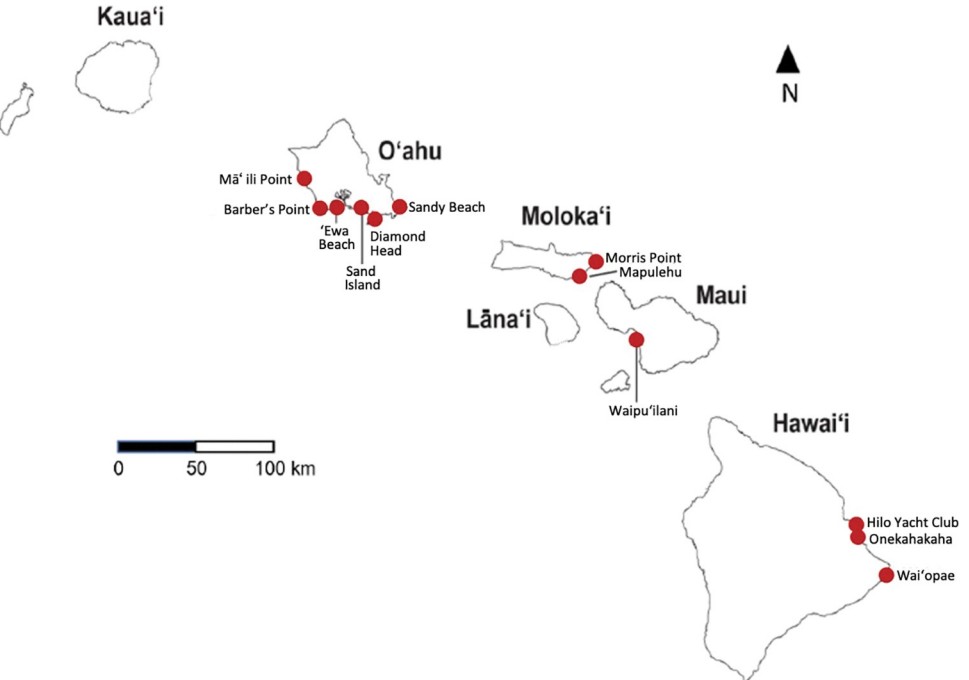

**Fig 1. Map of intertidal study sites.** Each of the 12 intertidal sites are represented with a red dot. Daily SSTs measurements from 2004–2019 were extracted for an area encompassing the entire Main Hawaiian Islands (~279,558 km$^2$). Mean daily SST was calculated from the MHI as a whole and daily SST measurements were also extracted for each intertidal site.

**Table 1. Sampling locations and frequency of sampling events.**

| Site | Island | # Quadrats | 2005 | 2006 | 2007 | No. of Sampling Events Per Year | 2016 | 2017 | 2018 | 2019 |
|---|---|---|---|---|---|---|---|---|---|---|
| Diamond Head | O'ahu | 957 | 3 | 1 | 2 | Eight-Year Gap in Data | 2 | 5 | 6 | 5 |
| Sand Island | O'ahu | 579 | 3 | 1 | 3 | | 3 | 4 | 5 | 1 |
| Barber's Point | O'ahu | 328 | 3 | 2 | 1 | | 7 | 3 | 2 | 0 |
| Sandy Beach | O'ahu | 129 | 0 | 1 | 1 | | 1 | 2 | 1 | 2 |
| 'Ewa Beach | O'ahu | 510 | 0 | 1 | 0 | | 3 | 8 | 3 | 7 |
| Mā'ili Point | O'ahu | 164 | 0 | 0 | 2[a] | | 0 | 3 | 0 | 3 |
| Waipu'ilani | Maui | 317 | 1 | 1 | 2 | | 2 | 2 | 2 | 0 |
| Wai'opae | Hawai'i | 110 | 0 | 0 | 4 | | 1 | 1 | 0 | 0 |
| Hilo Yacht Club | Hawai'i | 179 | 0 | 0 | 1 | | 0 | 1 | 2 | 2 |
| Onekahakaha | Hawai'i | 293 | 0 | 0 | 1 | | 3 | 4 | 3 | 3 |
| Morris Point | Moloka'i | 61 | 0 | 1 | 0 | | 1 | 0 | 0 | 0 |
| Mapulehu | Moloka'i | 126 | 0 | 1 | 0 | | 1 | 1 | 1 | 2 |

Twelve intertidal sites across Hawai'i sampled by the citizen science program Our Project in Hawai'i's Intertidal (OPIHI) with sampling events between 2005–2019. Total number of quadrats and the number of sampling events each year are provided. Gaps in sampling events are indicated in grey.

[a]Mā'ili Point was sampled once in 2007 and once in 2008. The two sampling events are shown together under 2007 for simplicity.

composition data were collected by the educational citizen-science program Our Project in Hawai'i's Intertidal (OPIHI, https://opihi.crdg.hawaii.edu/ [40]). From 2005–2019, OPIHI sampled 46 intertidal sites across the MHI (Table 1) [41]. Due to a hiatus in funding for the program, there is an eight-year gap in data collection from 2008–2015 [41] (Table 1). To test for an association between changes in SST and shifts in intertidal algal and invertebrate community composition, the sites selected for this study were those sampled before and after the eight-year gap in data collection with >60 quadrats surveyed per site overall. Data from eleven of the 12 sites selected for this study (apart from the Hilo Yacht Club) have been described in previous publications assessing spatial and temporal differences in algal and invertebrate community structure in relation to substrate [23,24]. This study reanalyzes the ecological data from these sites in a new analytical framework, with the inclusion of high-resolution SST data (see below), two years of newly collected intertidal ecological community data from 2018–2019, and the addition of the Hilo Yacht Club site.

## OPIHI sampling methods

Since 2004, the OPIHI program has paired professional marine scientists with secondary school students (grades 5–12) and their teachers to collect algal and invertebrate community-level intertidal data. Importantly, prior study has shown the citizen-science OPIHI data to be of comparable quality to that collected by experts in the field, but that the data are most appropriate for monitoring the more abundant species, rather than rare or cryptic species [21], therefore we focus on the more abundant taxa.

At each site, three to seven transects were laid perpendicular to the shoreline at varying lengths (8–30 m) and distances (2–10 m) apart from one another depending on the size of the intertidal bench, topography of the shore, and/or tidal height. The quadrat point-intercept method, described by Cox et al. [23], was then used to survey the algal and invertebrate community composition. Gridded 0.09 m² quadrats with 25 point-intercepts were set along the transect lines at regular intervals of 0.5 m, 1 m, or 2 m apart depending on location. Within each quadrat, any organism directly below each of the 25 point-intercepts was identified to the

lowest taxonomic level or functional group possible. Daily mean point-count abundance was then calculated for all identified organisms. Although the organisms in this study were sometimes identified at taxonomic levels (e.g., genus level) or functional groupings other than species, the term "taxa" is used to refer to all identified organismal groups in this study. Identification was based on OPIHI-developed identification guides [42] and field guides of Hawaiian algal [41] and invertebrate species [43]. All resulting data are made publicly available through the OPIHI website [40]. Permits were not required for the recording of these biological data, as no organisms were collected or harmed, and all beaches and shorelines in Hawaiʻi are in the public domain.

## Biological data processing

All taxa identified by OPIHI surveys were assessed during post-processing. Cryptic or rare species were grouped at higher taxonomic levels to be conservative with respect to identification, while morphologically distinct or dominant species were retained at the species level (S1 Table). For example, native *Laurencia* algal species in Hawaiʻi can be difficult to differentiate, so all *Laurencia* species documented by OPIHI surveys were assigned to the genus level (i.e., *Laurencia* spp.). Additionally, the microalgal conglomerate, turf algae, is referred to in this study as a single taxon for simplicity. In contrast, the invasive algae *Gracilaria salicornia* is a morphologically distinct and often-abundant alga found across the islands, so it was retained at the species level.

Dominant taxa both play key ecological roles in intertidal communities and are also the most confidently assessed with the OPIHI sampling methods [21]. Thus, for all taxa, overall mean point-count abundance was calculated, and the most abundant taxa were identified for the 12 intertidal sites separately and cumulatively. To capture changes in the overall community, the R package *vegan* [44,45] was then used to calculate Simpson's diversity index (1-D).

## Climatic data processing

Climatic data produced by the Operational Sea Surface Temperature and Sea Ice Analysis product (OSTIA), was obtained from the European Union's Earth Observation Program, Copernicus (https://marine.copernicus.eu). The OSTIA data are collected by both infrared and microwave satellite measurements, in addition to *in-situ* data from both moored and drifting buoys, to calculate daily foundational SST (SSTfnd) at a horizontal resolution of 1/20˚ (approx. 6 km$^2$) [46]. The OSTIA SSTfnd data were chosen for this study as they are free of diurnal variation (daytime warming or nocturnal cooling) and are considered to be a good representation of oceanic mixed layer temperatures (~1–10 m in depth) [46,47]. In this study, the terms SSTfnd and SST are used interchangeably.

Daily SST was acquired over a 16-year timeframe, from January 1st, 2004, through December 31st, 2019, for an area encompassing the MHI (279,558 km$^2$; Fig 1). Although the earliest OPIHI data in this study were collected in 2005, SST data from 2004 were obtained to provide insight into antecedent climatic conditions up to a year prior to each sampling date. The R package *ncdf4* [44,48] was then used to import SST data. To detect trends in SST across the MHI, mean daily SST was calculated for the full ocean area encompassing the MHI (Fig 1). To detect near-site variation in SST trends, daily SST was extracted for the 6 km$^2$ pixel nearest to each intertidal location.

Two SST satellite products are available from the OSTIA systems: a near real-time (NRT) SST product, which is made available 1–2 days after the sampling date, and a reprocessed (REP) SST product, which is released 1+ years behind NRT [49]. The REP data backfills gaps in data and increases confidence in temperature measurements [49]. At the time of data acquisition, OSTIA REP data were only available through December 31, 2018. To maximize the

amount of SST data available for analyses, the NRT data were used for the 2019 temperature measurements. To assess the compatibility of the NRT and REP SST products, Spearman's rank correlation tests were run for the entirety of the overlap between these two datasets within this study's timeframe. The two products were then plotted via a correlation plot, and prediction intervals were found for the mean, median, and maximum SST values at each intertidal site for the years with an overlap in NRT and REP data (2007–2018). Results from the Spearman's rank correlation test indicated the two satellite sources were strongly correlated with one another, with all *rho* values very close to 1 (S2 Table). The two product's variation in mean, minimum, and maximum SST values at each intertidal location fell between ±0.2–0.45˚C, indicating little variability between the two satellite data sources (S3 Table). Further, there was no general positive or negative relationship between the two temperature products, suggesting their variability is unlikely to systematically bias analyses.

## Statistical analyses

**Sea surface temperature analyses.** Although biological sampling spanned from 2005–2019, SST data from 2004–2019 were obtained and assessed; to allow for the inclusion of the possible influence of SSTs in the year prior (2004) to the first biological sampling dates (2005). To characterize change in SST over the course of the study's timeframe, the linear rate of mean daily SST change from 2004–2019 was calculated for the MHI (S4 Table). Linear rates of daily SST change from 2004–2019 were then calculated for the data pixel nearest or encompassing each of the 12 intertidal sites to explore possible differences in SST trends across the MHI (S4 Table). To capture non-linear trends in SST, a LOESS (locally estimated scatterplot smoothing) non-parametric curve fit was applied to the MHI and each intertidal site, compared with their linear regression models, and plotted with the R ggplot2 package (Figs 2 and 3) [50]. Further, to identify and elucidate the effect of any highly influential data points on warming trends (e.g., outlying extreme summer or winter SSTs), Cook's distance was calculated for the MHI's daily SST measurements. Observations with Cook's D values greater than 4/n (where n is the number of days sampled) were then removed and the linear rates of warming were again calculated and plotted (S1 and S2 Figs).

Because a gap in biological data collection occurred in the middle of the study's timeframe, from 2008–2015, it was important to understand how SST conditions may have varied between the sampling periods and over the eight-year gap. A series of two-tailed *t*-tests were used to determine if the mean, maximum, minimum, and variance of monthly SST across the MHI differed significantly from the first (2004–2011) and the second half (2012–2019) of the study's 16-year timeframe (S5 Table).

**Diversity and abundance analyses.** The R package *climwin* [51] was used to model the relationship between SST and the biological response variables (i.e., mean point-count abundance and diversity) for the 12 intertidal sites independently (i.e., site-level analyses) and collectively (i.e., MHI analyses). To better account for the influence of SST, *climwin* detects the time-windows preceding ecological sampling over which the biological responses are the most sensitive to SST [51]. For these analyses, daily SST data were collapsed into monthly summaries. The significance of the minimum, maximum, and mean monthly SST across the months within all possible time-windows and the goodness-of-fit among time-windows was then assessed by Akaike Information Criteria (AIC) [51]. Each model's optimal time-window was reported as the range of months, up to a year preceding ecological sampling, in which the long-term trends in SST (i.e., trends over the 16-year timeframe of the study) were most influential to the biological response. Two groups of analyses were conducted in *climwin* [51]. The first group consisted of site-level examinations, where each intertidal site was assessed

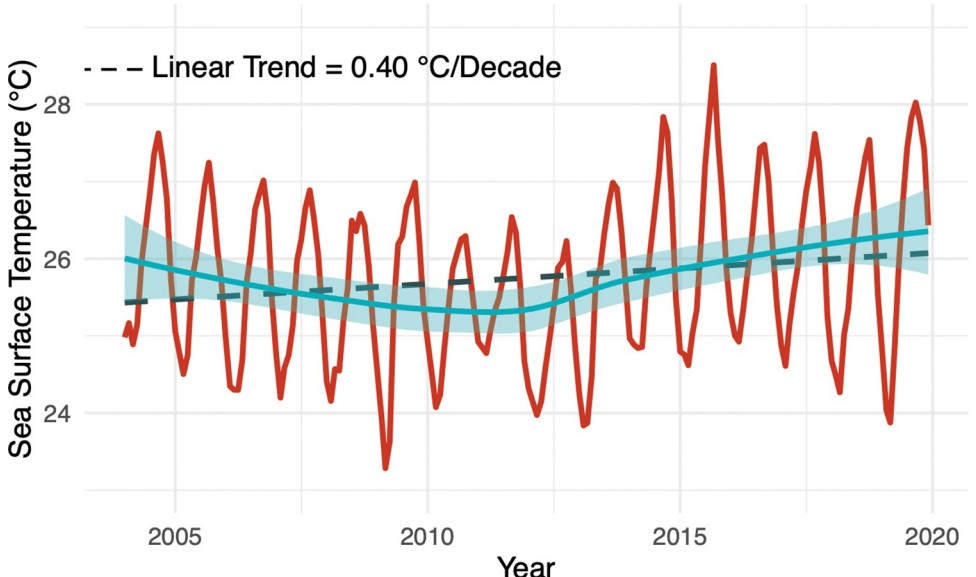

**Fig 2. Monthly SST for the MHI, 2004–2019.** Mean monthly SST measurements (red line), LOESS non-parametric curve fit (dark blue line), and 95% confidence interval (light blue band), and linear regression line (dashed grey line) for the Main Hawaiian Islands, 2004–2019. SST data were produced by the Operational Sea Surface Temperature and Sea Ice Analysis product (OSTIA), obtained from the European Union's Earth Observation Program, Copernicus (https://marine.copernicus.eu/).

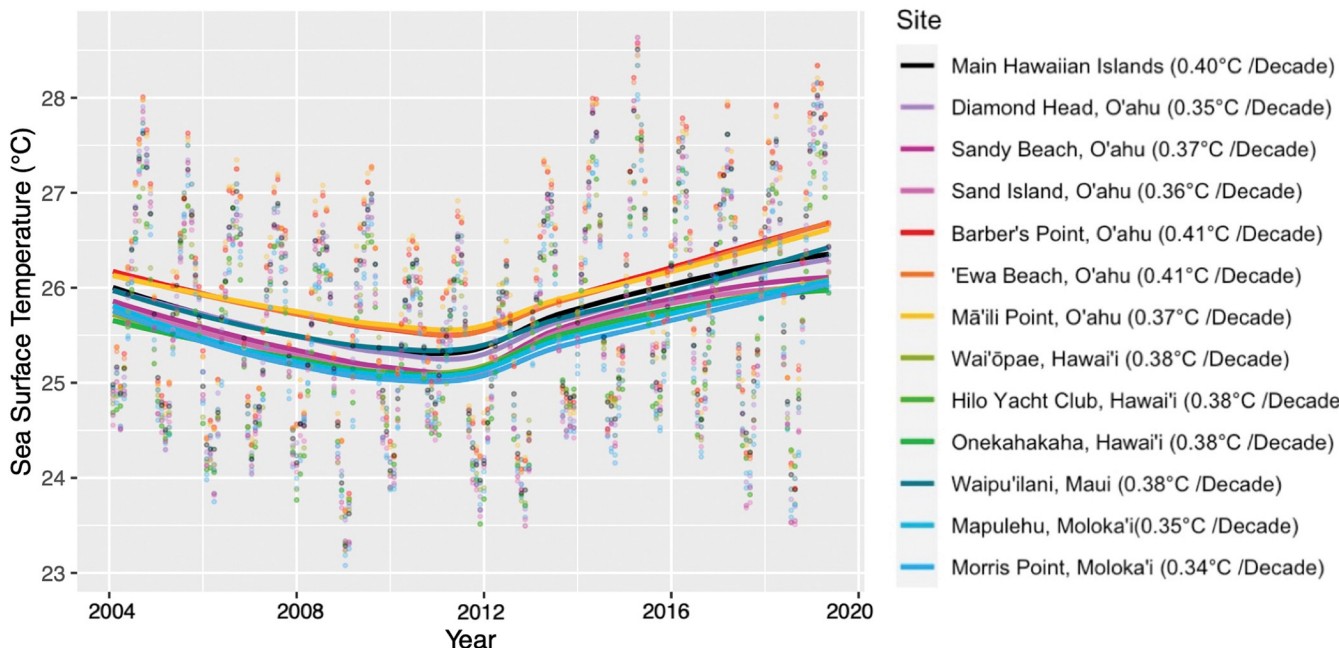

**Fig 3. SST Trends across the MHI and 12 intertidal study sites, 2004–2019.** LOESS Non-parametric curve fit of SST trends for the MHI collectively (black line) and the 12 intertidal study sites across the islands of Oʻahu, Hawaiʻi, Maui, and Molokaʻi from 2004–2019. All locations were found to have significant (p<0.01) linear warming trends ($R^2$ values: MHI = 0.029, Diamond Head = 0.020, Sandy Beach = 0.022, Sand Island = 0.021, Barber's Point, = 0.027, Ewa Beach, = 0.027, Māʻili Point = 0.022, Waiʻōpae = 0.034, Hilo Yacht Club, = 0.030, Onekahakaha = 0.030, Waipuʻilani = 0.026, Mapulehu = 0.022, Morris Point = 0.020).

independently, and the second of MHI analyses, in which the biological data from all intertidal sites were compiled and assessed together.

**Site-level analyses.** At each intertidal location, the five most abundant taxa were identified based on mean point-count abundance. The relationship between SST and daily mean point-count abundance was then independently assessed for each of the five taxa in *climwin* via negative binomial regression. A negative binomial regression was chosen for these analyses as it best accounted for the strong zero inflation observed in the data [52] while remaining compatible with the *climwin* program. A total of 60 negative binomial models were run, one for each of the five most abundant taxa at each of the 12 intertidal sites. Linear regression was then run in *climwin* to assess the relationship between SST and Simpson's diversity (1-D) at each site. A total of 12 linear regressions were run, one for each of the 12 intertidal sites.

**Main Hawaiian Islands analyses.** Among intertidal sites collectively, the 10 most abundant taxa were identified based on mean point-count abundance. The number of dominant taxa assessed was increased from five (at the site-level) to 10 (at the MHI-level) in order to more appropriately capture the range of taxa with potentially key ecological roles in the intertidal communities across the MHI. Additionally, this expansion accounted for site-level differences in community composition, as many taxa were only found as dominant community members in one site. The relationship between SST and daily mean point-count abundance of each taxon was then independently assessed in *climwin*, again via negative binomial regression. Spatial variation in SST between intertidal locations was accounted for in the model by pairing each site with the SST measurements for that location, rather than using an averaged MHI SST value for all sites. Linear regression was also used to assess how Simpson's diversity index (1-D) had cumulatively changed across all 12 intertidal sites.

A generalized dissimilarity model (GDM) was used to identify significant drivers of variability among the 12 intertidal sites. GDM's identify potentially influential predictor variables and provide the rate of compositional change that occurs across the environmental predictor variables (see below) [53,54]. Prior to running the GDM, a community matrix was generated wherein at each of the 12 intertidal sites, the mean point-count abundance of all identified taxa across all sampling events was calculated. Variation across all 12 intertidal sites was modeled as a function of five possible influencing variables (geographic distance among sites, 2004–2019 rate of SST warming, 2004–2011 SST mean, 2012–2019 SST mean, and mean SST for all sampling years: 2004–2019). Variables that significantly influenced these data were identified, and the percent deviance explained by the model was generated.

## Results

### Sea surface temperature trends

The linear regressions of daily SST across the MHI and for the 12 intertidal sites reveal warming trends ranging from 0.34–0.41°C Decade$^{-1}$ over this study's 16-year timeframe (S4 Table). LOESS non-parametric curve fit of daily SST across the MHI (Fig 2) and the individual intertidal sites (Fig 3) further demonstrates that the observed linear rate of SST warming was not a consistent trend across the study's timeframe. Rather, a cooling trend in SST was observed across the MHI from 2004–2011 (-0.62°C Decade$^{-1}$, p<0.01, $R^2$ = 0.29, SE = 0.08), and a warming trend in SST was observed from 2012–2019 (1.62°C Decade$^{-1}$, p<0.01, $R^2$ = 0.010, SE = 0.09). Maximum SSTs were significantly higher from 2012–2019 (mean = 26.3°C) than 2004–2011 (mean = 25.9°C, $t_{(190)}$ = 2.06, $p$ = 0.04, S5 Table). Additionally, mean SSTs were marginally significantly higher from 2012–2019 (mean = 25.9°C) than 2004–2011 (mean = 25.6°C, $t_{(190)}$ = 1.91, $p$ = 0.06, S5 Table). Minimum monthly SSTs and variance in monthly SSTs were not found to differ significantly between these time periods (S5 Table).

Trends in SST across sites were similar, with SSTs conditions at each intertidal location differing slightly from 2004–2019 (Fig 3, S5 Table). For example, Māʻili Point, ʻEwa Beach, and Barber's Point on Oʻahu tended to experience warmer SST conditions from 2004–2019 than the other sites and the MHI (Fig 3, S4 Table). Across the 16-year study period, Cook's D analysis identified 294 days as influential and pinpointed outlying high summer SSTs in 2004, 2005, 2015, 2017, and 2019 and outlying low winter SSTs in 2009, 2018, and 2019 (S1 Fig). When running the linear regression of daily SST across the MHI with these influential points removed, a slightly elevated warming rate was detected (0.48˚C Decade$^{-1}$; S3 Table).

## Site-level responses

Following biological data processing and filtering, a total of 116 algal and invertebrate taxa were identified across all 12 intertidal sites (S1 Table). The most abundant taxa often differed among the intertidal sites. The taxa which occurred the most frequently as one of the top five most abundant were turf algae (nine sites), crustose coralline algae (eight sites), *Padina* spp. (seven sites), *Acanthophora spicifera* (six sites), and brown crustose algae (five sites). Results of the *climwin* analyses revealed high variability in the relationship between SST and the daily mean point-count abundance for the five most abundant taxa across all sites and within each intertidal site (S6 Table). In particular, the strength and the direction of the relationship (reported as the % change in abundance per +0.1˚C in SST) and the most strongly influencing SST variable (i.e., minimum, maximum, or mean monthly SST) were highly variable (S6 Table). There were two instances where invasive algae were present as a dominant taxon within an intertidal site. Notably, at the site Onekahakaha on Hawaiʻi Island, the invasive algae *Gracilaria salicornia* was found to be highly dominant and increased in abundance by 35.1% for every 0.1˚C increase in minimum monthly SST (S6 Table). In contrast, while *Hypnea musciformis* was found to be a highly abundant taxon at Waipuʻilani, Maui, we found no significant relationship between its abundance and SST. Simpson's diversity index was significantly related to SST at eight of the 12 sites, with no significant relationship at Māʻili Point, Hilo Yacht Club, Morris Point, and Mapulehu (S7 Table).

## Main Hawaiian Islands responses

Across all intertidal sites, six of the 10 most abundant taxa assessed were found to have a significant relationship with increased SST (Fig 4, Table 2A). Five of the six taxa were algae (turf algae, *Padina* spp., *Acanthophora spicifera*, brown crustose algae, *Turbinaria ornata*) and only one was an invertebrate (*Dendropoma gregarium*; Table 2A). A positive increase in abundance of 21.8% for every 0.1˚C increase in the monthly SST statistic was identified for *Padina* spp. (95% CI = 13.1–31.4%), 27.0% for *Acanthophora spicifera* (95% CI = 16.3–39.9%), and 9.98% for *Dendropoma gregarium* (95% CI = 3.07–18.0%; Table 2A). In contrast, a decrease in abundance of 10.5% for every 0.1˚C increase in the monthly SST statistic was identified for turf algae (95% CI = 5.79–14.2%), 9.75% for brown crusting algae (95% CI = 2.28–17.4%), and 14.6% for *Turbinaria ornata* (95% CI = 5.23–23.8%; Table 2A). For five of these dominant taxa, with the exception of brown crustose algae, the time window over which they were the most sensitive to SST was six or more months wide (Table 2A, Fig 4). Additionally, the significant time-windows of these taxa occurred anywhere between the month of ecological sampling to 12 months preceding (Table 2A, Fig 4).

Although site-level diversity trends were highly variable, when Simpson's diversity index (1-D) was analyzed across all 12 sites, a significant 1.5% (95% CI = ± 1.0%) increase in diversity was identified for every 0.1˚C increase in mean monthly SST (F1, 134 = 9.66, p = 0.002; Fig 5, Table 2B). However, the model explains relatively little variation in the data ($R^2$ = 0.067), in

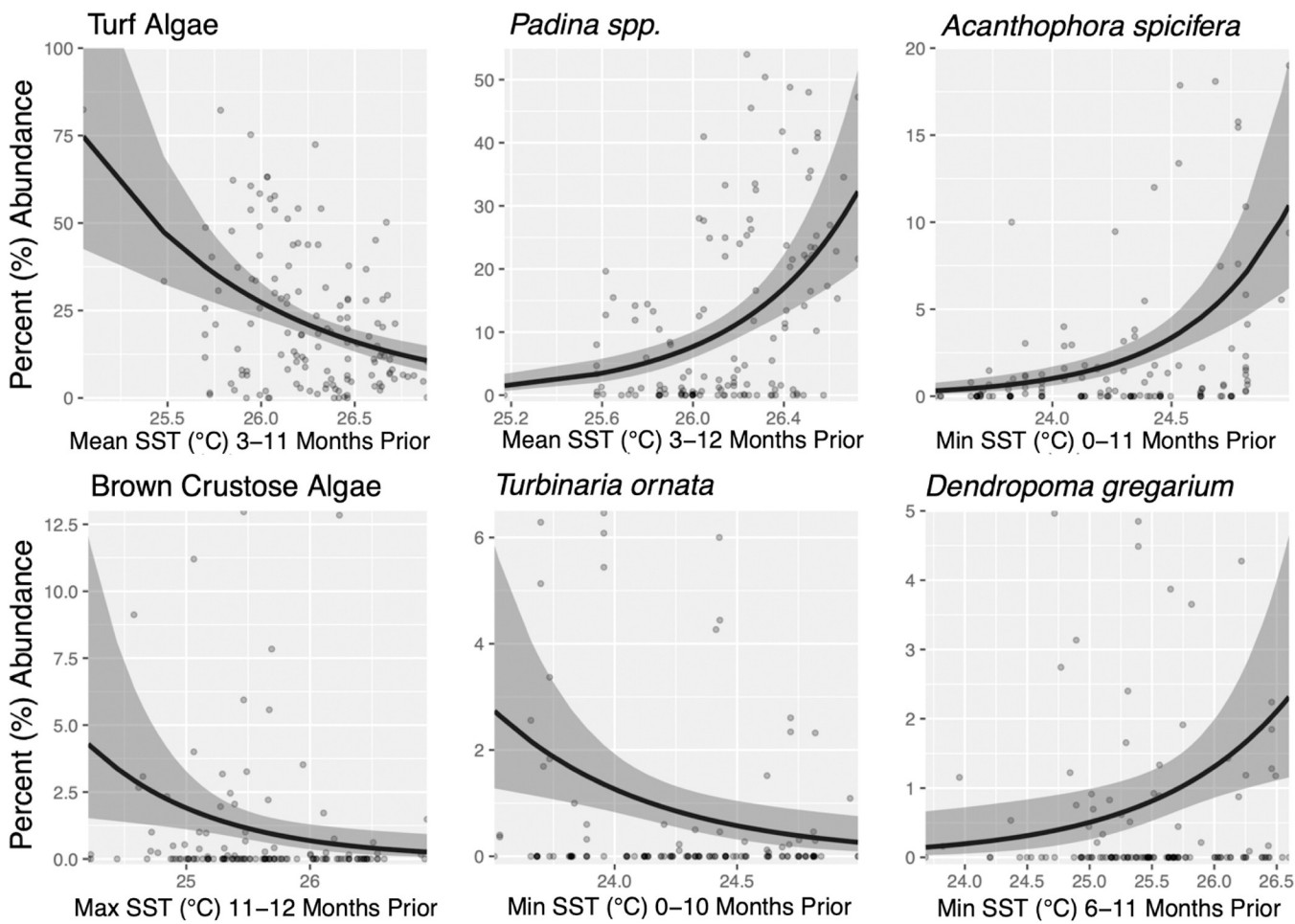

**Fig 4. Negative binomial relationship between sea surface temperature (SST) and daily mean point-count abundance for the six most abundant taxa with statistically significant (p<0.05) relationships across the 12 intertidal sites within the Main Hawaiian Islands from 2004–2019.** The negative binomial relationship modeled for each taxon (Table 2A) is indicated by the black line, 95% confidence intervals are in dark gray.

line with the observed high variability in site-level diversity trends. Similar to the highly abundant taxa, trends in SST over a wide time window (i.e., 1–10 months preceding sampling) were the most influential in describing the observed increase in diversity (Table 2B, Fig 5). This positive shift in Simpson's diversity index was driven by a decrease in the abundance of turf algae across all intertidal sites (Table 2B). Turf algae was the most abundant "taxon" across all sites, with an mean daily mean point-count abundance 1.75x greater than the nearest taxon (Table 2A; Fig 4), and is one of the top five most abundant taxa for nine out of 12 intertidal sites (S6 Table). Because of its dominance, the observed decrease in turf algae in relation to warming SST had a net effect of evening the abundance of all remaining taxa, increasing Simpson's diversity index as a result (Table 2B). When turf algae was removed from the MHI diversity analyses, a negative relationship was revealed between SST and Simpson's diversity index (-0.85% ± 0.077 per 0.1°C increase in mean monthly SST, $F_{1, 134} = 4.678$, $R^2 = 0.034$, p = 0.032, Table 2B), illustrating its fundamental role in the community composition at these intertidal sites.

The GDMs assessed community compositional dissimilarity across all 12 intertidal sites as a function of the five predictor variables (geographic distance among sites, rate of SST warming, 2004–2011 SST mean, 2012–2019 SST mean, and mean SST for all sampling years:

**Table 2. Model outputs for main Hawaiian Islands analyses.**

| A | Taxa | Mean Percent (%) Abundance | Climate Statistic | Time-Window (months prior) | Δ / +0.1°C | 95% CI | p-value |
|---|---|---|---|---|---|---|---|
| | Turf Algae | 22.2 | mean | 3–11 | (-) 10.5% | 5.79–14.2% | <0.001 |
| | *Padina* spp. | 12.6 | mean | 3–12 | (+) 21.8% | 13.1–31.4%% | <0.001 |
| | *Acanthophora spicifera* | 2.84 | min | 0–11 | (+) 27.0% | 16.27–39.9% | <0.001 |
| | Brown Crustose Algae | 1.21 | max | 11–12 | (-) 9.75% | 2.28–17.4% | 0.009 |
| | *Turbinaria ornata* | 0.99 | min | 0–10 | (-) 14.6% | 5.23–23.8% | 0.004 |
| | *Dendropoma gregarium* | 0.97 | min | 6–11 | (+) 9.98% | 3.07–18.0% | 0.008 |
| B | Simpson's Diversity | Climate Statistic | Time-Window (months prior) | Δ / +0.1°C | 95% CI | p-value | |
| | Main Hawaiian Islands | mean | 1–10 | (+) 1.55% | 0.56–2.54% | 0.003 | |
| | MHI w/o Turf Algae | min | 4–12 | (-) 0.85% | 0.39–2.26% | 0.03 | |

**(A)** The negative binomial relationship between sea surface temperature (SST) and daily mean point-count abundance for the six most abundant taxa with significant relationships across the 12 intertidal sites within the Main Hawaiian Islands. For each taxon, six metrics are provided: 1) daily mean point-count abundance, 2) the most influential climate statistic (mean SST, maximum SST, or minimum SST), 3) the time-window (reported as months preceding ecological sampling) most influential for describing observed trends, 4) the percent change in abundance per 0.1°C increase in monthly SST statistic, 5) 95% confidence intervals, and 6) p-values. **(B)** The linear relationship between SST and Simpson's Diversity across all 12 sites together, both with and without turf algae included in the analyses. Five metrics are provided: 1) the most influential climate statistic (mean SST, maximum SST, or minimum SST), 2) the time-window (reported as months preceding ecological sampling) most influential for describing observed trends, 3) the percent change in abundance per 0.1°C increase in monthly SST statistic, 4) 95% confidence intervals, and 5) p-values.

2004–2019). The rate of SST warming (p = 0.48) and mean SST from the second half of this study's timeframe (2012–2019; p = 0.92) were not significantly associated with community dissimilarity among sites and were excluded from the full model results reported below. Results from the GDM with geographic distance, mean SST from the first half of this study's

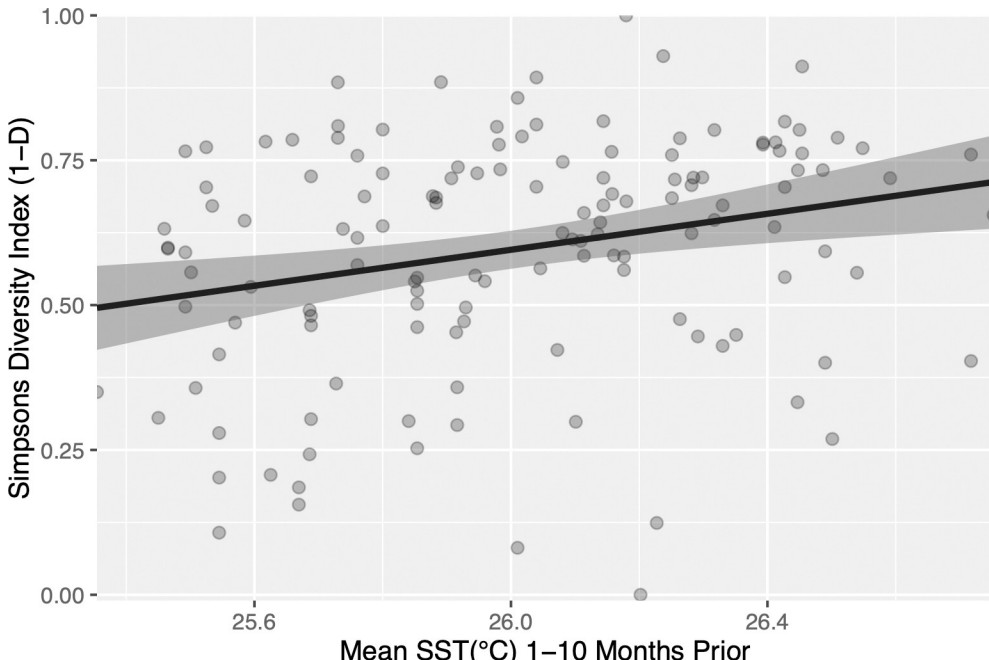

**Fig 5. Linear relationship between Simpson's Diversity and SST.** The linear relationship between Simpson's Diversity and mean sea surface temperature (SST) 1–10 months prior to the sampling date for all 12 intertidal sites collectively across the islands of Oʻahu, Hawaiʻi, Maui, and Molokaʻi from 2004–2019. The linear regression is indicated by the black line, 95% confidence interval in dark gray.

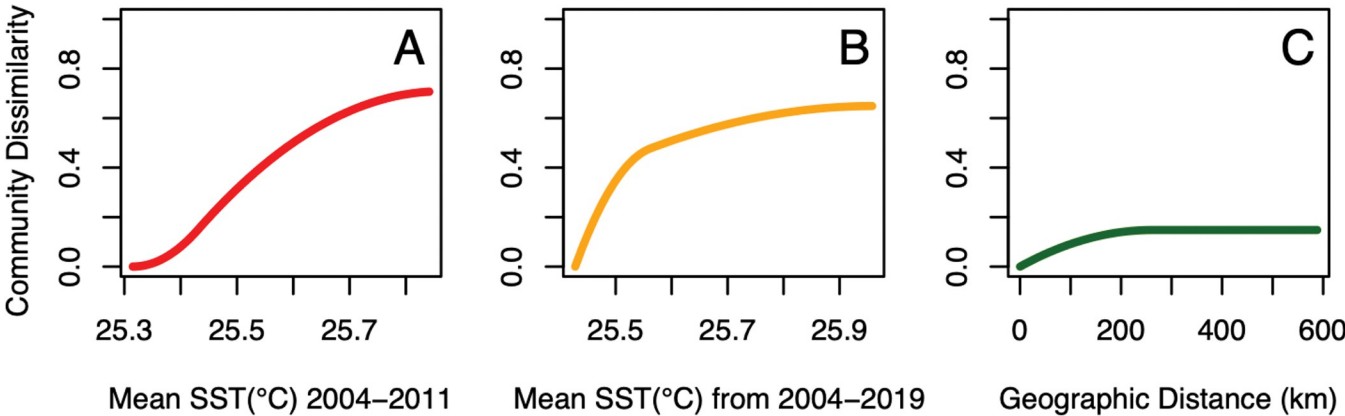

**Fig 6. Generalized dissimilarity model-fitted partial regressions.** Generalized dissimilarity model-fitted partial regressions for all variables found to significantly explain community dissimilarity. Each curve's maximum height represents the total amount of community dissimilarity associated with the variable and the slope of the lines represents the rate at which community dissimilarity increases along the variable's gradient. Model results explained 45.1% of the deviance in community composition between sites (p<0.05). **(A)** Mean SST from the first half of this study's timeframe (2004–2011) was the most influential variable for explaining dissimilarity between intertidal communities (magnitude = 65.4, p<0.05; red line), followed by **(B)** mean SST for the entire study period (2004–2019; magnitude = 24.6, p = 0.04; orange line), and then by **(C)** geographic distance (magnitude = 1.94, p<0.05; green line).

timeframe (2004–2011), and mean SST for the entire study period (2004–2019) as predictor variables explained 45.1% of the deviance in community composition among sites (p<0.05; Fig 6). Model results provide the magnitude of each predictor variable as a measure of how influential each variable is for explaining community dissimilarity. Mean SST from the first half of this study's timeframe (2004–2011) was found to be the most influential variable for explaining dissimilarity among intertidal communities (magnitude = 65.4, p<0.05), followed by mean SST for the entire study period (2004–2019; magnitude = 24.6, p = 0.04), and geographic distance (magnitude = 1.94, p<0.05).

## Discussion

### Sea surface temperature warming trends

Outputs from two major global climate models assessing SST from 1976–2099 at current greenhouse gas emission rates predict high rates of SST warming across Hawai'i through the end of the 21$^{st}$ century [3]. We found that across the MHI and for the 12 intertidal sites, the observed linear rates of SST warming closely aligned with predicted rates of warming for the end of the century (predicted: 0.39°C Decade$^{-1}$ from 2070–2100 [3], observed: 0.34–0.41°C Decade$^{-1}$, S4 Table). However, the LOESS non-parametric curve fits for the MHI and for all intertidal sites reveals that the rate of warming observed over the course of the study timeframe was largely influenced by rapidly increasing SST from 2012–2019 (1.62°C Decade$^{-1}$). Additionally, SSTs from 2012–2019 were found to have greater maximum and mean monthly SST (S5 Table). This change in SST trends from 2004–2011 to 2012–2019 could be a signal of large-scale oceanographic patterns, such as PDO (Pacific Decadal Oscillations) or ENSO (El Niño/ Southern Oscillations) [55,56]; although the relatively limited timeframe of the current study makes it difficult to adequately address their possible role(s). Additionally, two marine heatwaves occurred over the course of this study (2014–2015 and 2019) that were both drivers of moderate to severe coral bleaching events across the archipelago [57,58]. Because the detected warming trend persisted when re-calculated without outliers (S2 Fig), the changes in SST observed over the course of this study are unlikely to be driven solely by these anomalous SST events.

## Site-level responses and community dissimilarity in relationship to SST

Previous studies have demonstrated that Hawaiian intertidal communities can vary greatly from one another, with interactions between local biotic and abiotic factors driving differences in community composition that are often difficult to predict [22,23,27]. This study's site-level analyses support these prior general findings and further indicate high variability in community response to SST among intertidal sites (S6 and S7 Tables). In addition, the response of Simpson's diversity index to changing SST also differed among sites (S7 Table). While the abundance of the invasive algae, *Gracilaria salicornia* was found to have a positive relationship to SST at the site Onekahakaha on Hawai'i Island (S6 Table), additional investigation is required to understand how this relationship may be expressed across intertidal sites with differing local biotic and abiotic factors, as this invasive species may continue to extend its range. Although several of the same taxa were found to be abundant across multiple intertidal communities (e.g., turf algae, *Padina* spp., *Acanthophora spicifera*), their responses to warming often varied in both magnitude and direction between sites (S6 Table), underscoring the complexity of the drivers of change in these communities.

While warming trends were similar across the MHI and among the 12 sites from 2004–2019, variability in SST conditions across sites (Fig 3, S5 Table) significantly described differences in community composition (Fig 6). Across the 12 intertidal sites, variation in SST explained 45.1% of the dissimilarity in community composition (Fig 6). Mean antecedent SST from the first half of this study's timeframe (2004–2011) was the most influential variable for explaining dissimilarity among intertidal communities (magnitude = 65.4, p<0.05), followed by mean SST for the entire study period (2004–2019; magnitude = 24.6, p = 0.04), and then by geographic distance (magnitude = 1.94, p<0.05; Fig 6). More recent SST conditions (2012–2019) and the rate of warming across the entire timeframe of the study were not significantly associated with community dissimilarity. These results point to the lasting influence of local antecedent SSTs on shaping intertidal community composition and suggest that even slight differences in antecedent SST conditions are sufficient to affect community structure and drive dissimilarity among locations (Fig 6).

## Main Hawaiian Islands responses to antecedent SST conditions

Across many coastal regions of the globe, shifts in SST have been found to strongly influence community composition at large spatial scales [10,34,39,59]. Multiple factors that covary with SST may be driving this, as subtle differences in SST are known to influence larval success and dispersal [6,60] and impact interspecific interactions [7–9]. At the site-level, the responses of highly abundant taxa and community diversity to SST change were found to be highly variable. However, clear responses in abundance and diversity were detected across the MHI (Table 2; Figs 4 and 5). At the MHI scale, significant relationships between SST, diversity, and six highly abundant taxa (in descending order of abundance: turf algae, *Padina* spp., *Acanthophora spicifera*, brown crustose algae, *Turbinaria ornata*, and *Dendropoma gregarium*; Table 2, Figs 4 and 5) were identified. These six abundant taxa play key ecological roles within Hawaiian intertidal communities as primary producers (i.e., brown crustose algae, *Padina* spp., turf algae, *Turbinaria ornata*) [61], erosion control (*Dendropoma gregarium*) [62], and invasive threats (*Acanthophora spicifera*) [63]. Changes in the abundance of these often-dominant community members largely explained the increase in diversity in response to SST warming across the MHI (Table 2). In particular, the decrease of turf algae, which had an mean abundance ~1.75x greater than the second most abundant taxon, resulted in an overall increase in taxonomic evenness (i.e., the abundances of other taxa became more similar), and in turn an increase in Simpson's diversity index (Table 2B). Considering that increases in turf algae are often

associated with decreased coral cover following bleaching events in subtidal communities [64], the decreased abundance of turf algae across the MHI with warming SSTs is perhaps unexpected. Turf algae have been found to be opportunistic in warming SST, with thermal thresholds that surpass those of their competitors and allow them to proliferate where other taxa may experience declines [65]. These findings may suggest turf algae is approaching the upper limits of its acclimation capacity to warming [66]. Additionally, shifts in the abundance of all six highly abundant taxa could be due to additional biotic or abiotic variables (e.g., grazing [25], anthropogenic disturbance [28], nutrient influx [27]) interacting with the effects of SST, although further research is needed to address these hypotheses.

Wide time-windows (i.e., 6+ months) occurred between the month of ecological sampling to 12 months preceding and were the most influential for describing changes to diversity and the abundance for five out of six highly dominant taxa across the MHI (the exception being brown crustose algae, Table 2, Figs 4 and 5). These results highlight antecedent SSTs over the 6–12 months preceding ecological sampling as significant drivers of change within Hawaiian intertidal communities (Table 2, Figs 4 and 5) and further complement the finding that antecedent SST conditions have a lasting influence on community dissimilarity among intertidal sites (Fig 6). Previous studies have identified narrower time-windows (i.e., 4–5 weeks preceding sampling [30–32] and inter-annual seasonality [29]) as influential for algal community biomass within tropical subtidal and reef habitats. The temporal scales of these studies were restricted, each with sampling periods of less than three years [29–32]. While the trends associated with these narrower time-windows describe interannual fluctuations in intertidal algal biomass [29–32], they may be too limited to define how these systems are responding to longer-term changes in SST. By determining the significance of long-term warming SST trends over the 6–12 months preceding ecological sampling, this study further highlights the need for wider sampling time frames of intertidal communities.

## Future studies and implications

Large-scale oceanographic patterns, such as the PDO, are associated with long-term fluctuations in SST variability, with periodicities often ranging from 15–25 or 50–70 years in length [55]. Due to the relatively limited timeframe of this study, further research is required to extract the influence of these climatic phenomena on SST trends across Hawaiʻi within the 16-years assessed in this study and beyond. Additionally, SST warming trends are known to be localized or dampened due to influences such as ocean currents, surface winds [67], and/or groundwater influx [68], and the degree of thermal stress on intertidal organisms can be influenced by additional factors such as air temperature, solar radiation, wind, and wave/tidal fluctuations [69]. Future research incorporating these abiotic factors (among others) would be valuable. The significant influence of subtle differences in antecedent SST among intertidal sites on community dissimilarity (Fig 6) suggests that future intertidal studies should also be conducted at the finest temporal resolution possible. In addition, satellite-derived data, such as the OSTIA products used in this study [46], are an excellent resource for obtaining longer-term climatic data at a relatively fine spatial scale ($\sim 6$ km$^2$) when continual in-situ monitoring is not a possibility. To continue to better account for the role of SST and additional climatic variables on intertidal communities, it would be valuable to expand investigations to encompass longer timeframes (i.e., multiple years [33] to decades-long [10,34,35]) and to also assess the role of SST changes over a wider range of antecedent time-windows [51].

Continued funding and support for citizen and community-driven science initiatives, such as OPIHI [40], can provide a valuable, long-term source of high-quality intertidal ecological community data for sites across the state [21]. Future studies that build on our methods to

explore the lagging influence of SST on intertidal community change, specific functional groups, and invasive taxa are important subsequent steps in this line of inquiry. In addition to temporal variation, as our results and the results of other recent studies have found, Hawaiian intertidal ecological communities can vary significantly across small spatial scales [23,24] and in response to multiple biotic and abiotic ecological determinants through time. For this reason, whenever feasible, we recommend future analyses of the Hawaiian intertidal be conducted at multiple spatial and temporal scales to maximize confidence in detected trends.

We acknowledge that, while our analyses assessing the relationship between SST and abundance or diversity have an exploratory nature, they provide a first foundation of insights into the influence of SST on Hawaiian intertidal communities, and we offer the following suggestions to further address these hypothesized relationships. Due to the detected significance of multi-year trends in antecedent SST on Hawaiian intertidal community structure, we predict continued lagging shifts in community composition will be observed as rates of warming continue to increase in both the near and distant future [3]. In temperate intertidal systems along the coastal United States [10], South America [33], and Europe [70,71], community shifts across large spatial scales in response to ocean warming have already been documented. Within these regions, poleward shifts in species' range and abundance have been observed [10,33], with populations migrating towards cooler waters as thermal stress increases. At the same time, the successful invasion of many introduced marine organisms has been linked to their greater thermal tolerances and acclimation potential [64,66]. For example, differences in thermal tolerances have been linked to the invasion and proliferation of algae on coral reefs following marine heatwaves [64]. Due to the MHI limited latitudinal range (i.e., only ~3˚ from Hawai'i Island to Kaua'i) and extreme geographic isolation, there is little opportunity for intertidal populations to migrate poleward in the island chain (most of the small islands and atolls to the northwest in Papahānaumokuākea do not provide similar intertidal habitat to the MHI), increasing the potential to reach the limit of their thermal tolerances [66]. Ultimately, this could result in severe population declines for some native species and opportunity for warm water tolerant species to proliferate [66].

Although rates of warming were similar across the 12 sites assessed in the MHI, differences in SST among intertidal sites were detected (Fig 3). Previous studies have described invertebrate and algal responses to ocean warming in similarly complex thermal environments [59], resulting in a 'mosaic' of community responses, where species turnover increases and community stability decreases as thermal stress rises [33,59,72]. Results of this study highlight how slight variation in SST across the Hawaiian Islands can play a key role in shaping intertidal community composition (Figs 3 and 6). Particular consideration should be given to Hawaiian intertidal sites experiencing elevated SST conditions and/or rates of warming, as they may be the most likely to display large shifts in community composition, but that time-lags on the order of months to years are to be expected before these changes occur.

## Conclusion

The SST trends observed over this study's timeframe capture periods of rapid SST warming for the Hawaiian intertidal (particularly 2012–2019; Fig 2), at or exceeding expected rates for the end of the 21st century [3,56]. Although site-level responses to SST warming in regard to abundance and diversity were highly variable (S6 Table), at the MHI scale, antecedent SSTs over the several months to a year preceding ecological sampling were identified as significant drivers of change within these intertidal communities (Table 2, Figs 4 and 5). Additionally, site-level variation in mean SST conditions (primarily in 2004–2011 and secondarily in 2004–2019) significantly described overall community dissimilarity among intertidal sites (Fig 6).

These results highlight that although Hawaiian intertidal algal and invertebrate communities are highly variable among sites and across islands, both longer-term trends in SST and antecedent SST up to a year prior can have a lasting influence on shaping intertidal communities across the MHI. Future research and management are encouraged to consider these lagging effects of SST on intertidal communities, as short-term fluctuations in these systems may not be representative of longer-term community responses to changes in SST.

## Supporting information

**S1 Fig. Plot of influential SST data points from daily SST measurements of the MHI.** Influential data points (indicated as red dots) were identified via Cook's Distance, and all observations with a Cook's D value greater than $4/n$ (indicated as the red line, where $n$ is the number of observations) are highlighted in red. For comparative purposes, Cook's D plot (**B**) is stacked beneath a plot of daily SST values (**A**). Influential data points were identified as the highest summer SST highs in 2004, 2005, 2015, 2017, and 2019 and the lowest winter SST lows in 2009, 2018, and 2019.
(TIF)

**S2 Fig. A side-by-side comparison of the linear SST warming trends for the MHI with and without influential data points (as indicated by Cook's D analysis).** The original rate of warming with daily SST on the left, and the plot on the right has influential data points removed. Without the influential points (0.48˚C Decade$^{-1}$), the rate of warming is slightly greater than with them included (0.40˚C Decade$^{-1}$).
(TIF)

**S1 Table. (A-E)** Classifications of all algae and invertebrates documented in the Our Project in Hawai'i's Intertidal (OPIHI) database. Highly cryptic or extremely rare species were grouped at higher taxonomic levels to be conservative with respect to identification, while easily identifiable or highly abundant species were retained at the species level. Additionally, taxa were separated into organism type, group, and invasive classification (i.e., native, native-dominant, non-native invasive, and unknown). 'Native-dominant' categorizes algae or invertebrates which are native to the Hawaiian Islands but considered highly noxious and have been recognized for their potential to dominate Hawaiian habitats. The 'non-native invasive' classification refers to organisms often with high tolerances to disturbance and thermal variation that introduced to the Hawaiian Islands and are currently listed as invasive species by Hawai'i's Division of Aquatic Resources (DAR 2020). The classification, 'native' is for organisms introduced the Hawaiian Islands which are not known to be noxious or dominant in Hawaiian intertidal habitats. Species with ambiguous identifications were classified as 'unknown'.
(PDF)

**S2 Table. Spearman's rank correlation test to assess the compatibility of NRT and REP satellite data sources at each intertidal location from 2005–2018.**
(PNG)

**S3 Table. Prediction intervals for mean, minimum, and maximum SST values of NRT and REP satellite data sources at each intertidal location from 2005–2018.**
(PDF)

**S4 Table. Linear regression of SST from 2004–2019 as a function of time for the MHI and the 12 individual sites across the Main Hawaiian Islands.**
(TIF)

**S5 Table. (A)** Summary statistics for mean yearly SST across the MHI from 2004–2019. Dates in gray are from the first sampling period (2004–2011) and dates in white are from the second sampling period (2012–2019). **(B)** Results from two-tailed *t*-test comparing monthly sea surface temperature (SST) statistics from 2004–2011 to 2012–2019 for the Main Hawaiian Islands.
(PDF)

**S6 Table. The relationship between SST and point-count abundance for each of the five most abundant taxa within 12 intertidal sites across the Main Hawaiian Islands (MHI).** Additionally, the relationship between SST and point count abundance was also assessed for the 10 most abundant taxa among all 12 sites cumulatively. All analyses were conducted in *climwin* via negative binomial regression. Grouped by taxa, only statistically significant relationships are shown.
(PDF)

**S7 Table. The relationship between SST and Simpson's diversity (1-D) among 12 intertidal within the Main Hawaiian Islands, assessed via linear regression in *climwin*.** Out of 12 intertidal sites assessed, only the eight found to have a statistically significant relationship between diversity and SST warming are shown.
(PDF)

## Acknowledgments

We would like to thank the Our Project In Hawai'i's Intertidal (OPIHI), as this work would not be possible without the efforts of the many teachers, students, and volunteers involved in the program. Thank you to Steven Colbert and Tracy Weigner, who shared their knowledge and expertise throughout the development and implementation of this study. Additionally, we would like to thank Noell Hadad, Catherine McTighe, and Megan Nakamoto for their support in post-processing biological data.

## Author Contributions

**Conceptualization:** Rebecca J. Ward, T. Erin Cox, Anuschka Faucci, Florybeth Flores La Valle, Joanna Philippoff, Jessica L. B. Schaefer, Ian M. Ware, Matthew L. Knope.

**Data curation:** Rebecca J. Ward.

**Formal analysis:** Rebecca J. Ward.

**Investigation:** Rebecca J. Ward.

**Methodology:** Rebecca J. Ward, T. Erin Cox, Anuschka Faucci, Florybeth Flores La Valle, Joanna Philippoff, Jessica L. B. Schaefer, Ian M. Ware, Matthew L. Knope.

**Project administration:** Rebecca J. Ward.

**Resources:** Rebecca J. Ward.

**Supervision:** Matthew L. Knope.

**Visualization:** Rebecca J. Ward.

**Writing – original draft:** Rebecca J. Ward.

**Writing – review & editing:** Rebecca J. Ward, T. Erin Cox, Anuschka Faucci, Florybeth Flores La Valle, Joanna Philippoff, Jessica L. B. Schaefer, Ian M. Ware, Matthew L. Knope.

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
