## [Decision Letter · Decision Letter 0]

2 Jan 2023

PONE-D-22-26540Spatial Variation and Antecedent Sea Surface Temperature Conditions Influence Hawaiian Intertidal Community StructurePLOS ONE

Dear Dr. Ward,

Thank you for submitting your manuscript to PLOS ONE. After careful consideration, we feel that it has merit but does not fully meet PLOS ONE’s publication criteria as it currently stands. Therefore, we invite you to submit a revised version of the manuscript that addresses the points raised during the review process.

Upon the Reviewer recommendations and my own assessment of the manuscript, major and mandatory revisions are required before this manuscript may become acceptable for publication with PLOS ONE.  Several figures/tables are missing from the manuscript, and more detailed information about underlying data and results are required to facilitate interpretation of the results. The Reviewer comments provide further recommendations on the statistical analysis to ensure the robustness of the findings to distinguish long-term trends from multi-decadal variability.

We look forward to receiving your revised manuscript.

Kind regards,

Caroline Ummenhofer

Academic Editor

PLOS ONE

Journal Requirements:

" ext-link-type="uri" xlink:type="simple">https://journals.plos.org/plosone/s/file?id=ba62/PLOSOne_formatting_sample_title_authors_affiliations.pdf"

3. "We note that Figure (1) in your submission contain map images which may be copyrighted. All PLOS content is published under the Creative Commons Attribution License (CC BY 4.0), which means that the manuscript, images, and Supporting Information files will be freely available online, and any third party is permitted to access, download, copy, distribute, and use these materials in any way, even commercially, with proper attribution. For these reasons, we cannot publish previously copyrighted maps or satellite images created using proprietary data, such as Google software (Google Maps, Street View, and Earth). For more information, see our copyright guidelines: http://journals.plos.org/plosone/s/licenses-and-copyright.

1. You may seek permission from the original copyright holder of Figure (1) to publish the content specifically under the CC BY 4.0 license.  

Natural Earth (public domain): " ext-link-type="uri" xlink:type="simple">http://www.naturalearthdata.com/"

Reviewers' comments:

Reviewer's Responses to Questions

**Comments to the Author**

1. Is the manuscript technically sound, and do the data support the conclusions?

Reviewer #1: Yes

Reviewer #2: Partly

Reviewer #3: Yes

2. Has the statistical analysis been performed appropriately and rigorously? 

Reviewer #1: Yes

Reviewer #2: I Don't Know

Reviewer #3: Yes

3. Have the authors made all data underlying the findings in their manuscript fully available?

Reviewer #1: Yes

Reviewer #2: No

Reviewer #3: Yes

4. Is the manuscript presented in an intelligible fashion and written in standard English?

Reviewer #1: Yes

Reviewer #2: Yes

Reviewer #3: Yes

5. Review Comments to the Author

Reviewer #1: This was a very nicely written, pertinent and interesting paper and i enjoyed reading it. There are just a few formatting errors though that need correction as follows (indicated by line number):

30 - km2, should be a superscript

All though the paper you are inconsistent in the rule as to use of numerals or words for numbers. The usual rule is the use words for numbers up to nine and then numerals for 10 and upwards. You do generally follow this, but repeatedly talk about twelve (should be 12) sites (lines 30 151, 193 etc many times) and sixteen (should be 16) years (e.g. lines 206 266 and many other places). You also have ten not 10 on lines 296, 298, but correctly 12 on line 513 for example! In the reverse way you have several number under 10 on lines 344 and these should be given as words! This must at least be consistent though entire paper whatever rule you follow.

422, also 446, 452,575 surely this should read '21st century'

650 species name must have a lower case

753, 787 Do not capitalise the words in the titles of journal articles ( even if this was done in the original journal, as some do do)

766 there are 2 Smith refs here, the second one needs a new line and new number

776, 788 you need to abbreviate this journal title for consistency

Reviewer #2: This paper reports a study examining changes in Hawaiian shoreline communities in relation to sea surface temperature. The paper potentially offers new and useful information on the ecology of these systems. However, in its current form, the paper is not reviewable. Although the core of the paper is the abundance of shoreline species, key figures with these data are missing in my copy, specifically Supplementary Tables 6 and 7 and Figure 6.

Beyond missing figures, most figures in the paper only provide statistical summary lines/curves rather than the biological data themselves. It is essential that the underlying data also be provided so that the reader can assess the fits to the data. This is especially true of Figure 4, which offers relationships with the species abundance data, but is also true for Figs. 3 and 5.

More information in Figure 2 is also needed. Specifically, a linear temporal trend is provided, but visually the temporal patterns appears to exhibit an oscillatory pattern across the duration of the data (beyond the annual cycles). Perhaps there is a signal of a relationship to some large-scale oceanographic pattern such as the PDO that can be extracted, or perhaps using a non-linear regression with an annual cosine function, an unspecified cosine function on the order of 20 yr (make the period a variable to be fitted) and a linear function would clarify the magnitude of a potential trend. The concern is that a longer-period oscillation can look like a long-term trend and the pattern in the data indicates a distinctly non-linear trajectory through time. At the least, a non-parametric curve fit (LOESS, DWLS or similar) would be beneficial.

Reviewer #3: This is a very timely and interesting study that does very well in incorporating citizen science. I found the methods to be rigorous and well explained, findings and interpretations to be justified, and the manuscript to be very well written. I normally have more comments and feedback to authors, but in this case there's not much more to say. I look forward to seeing this article published.

6. PLOS authors have the option to publish the peer review history of their article (what does this mean?). If published, this will include your full peer review and any attached files.

Reviewer #1: **Yes: **Charles L Griffiths

Reviewer #2: No

Reviewer #3: No

---

## [Author Response · Author response to Decision Letter 0]

21 Feb 2023

Dear Dr. Ummerhofer, 

Thank you for serving as the handling editor and for reviewing our manuscript. Please see below for a point-by-point response to all comments. In short, we made all changes and clarifications requested. Changes are provided below, following each numbered journal requirement. Additionally, for your convenience any edited figures/tables and their legends are included in the Response to Reviewer's letter, which is included at the end of the resubmitted Manuscript Draft PDF.

Journal Requirements

https://journals.plos.org/plosone/s/file?id=ba62/PLOSOne_formatting_sample_title_authors_affiliations.pdf"

We completed a review of all style requirements. File names were edited to include the file type at the end of the name (i.e., previously: “Fig1”, corrected: “Fig1.tiff”). 

Completed. A brief statement was added to the end of the Methods section OPIHI Sampling Methods which states: “Permits were not required for the recording of these biological data, as no organisms were collected or harmed, and all beaches and shorelines in Hawaiʻi are in the public domain.”

3. "We note that Figure (1) in your submission contain map images which may be copyrighted. All PLOS content is published under the Creative Commons Attribution License (CC BY 4.0), which means that the manuscript, images, and Supporting Information files will be freely available online, and any third party is permitted to access, download, copy, distribute, and use these materials in any way, even commercially, with proper attribution. For these reasons, we cannot publish previously copyrighted maps or satellite images created using proprietary data, such as Google software (Google Maps, Street View, and Earth). For more information, see our copyright guidelines: http://journals.plos.org/plosone/s/licenses-and-copyright.

We require you to either (1) present written permission from the copyright holder to publish these figures specifically under the CC BY 4.0 license, or (2) remove the figures from your submission. 

 We have received written permission from the copyright holder of Figure 1 (Dean Lodes, Director of Marketing and Publication Services with the Curriculum Research Development Group at the University of Hawai‘i – Mānoa) to publish the image under the CC BY 4.0 license. The completed content permission form will be uploaded as an “Other” file with the resubmission.

Reviewer #1

1. This was a very nicely written, pertinent and interesting paper and i enjoyed reading it. There are just a few formatting errors though that need correction as follows (indicated by line number):

 Thank you very kindly for your positive comments and helpful edits All suggested changes have been made to the manuscript and are reflected in the updated draft and document with tracked edits. 

2. 30 - km2, should be a superscript 

 Corrected

3. All though the paper you are inconsistent in the rule as to use of numerals or words for numbers. The usual rule is the use words for numbers up to nine and then numerals for 10 and upwards. You do generally follow this, but repeatedly talk about twelve (should be 12) sites (lines 30 151, 193 etc many times) and sixteen (should be 16) years (e.g. lines 206 266 and many other places). You also have ten not 10 on lines 296, 298, but correctly 12 on line 513 for example! In the reverse way you have several number under 10 on lines 344 and these should be given as words! This must at least be consistent though entire paper whatever rule you follow. 

 Corrected throughout the manuscript

4. 422, also 446, 452,575 surely this should read '21st century' 

 Corrected

5. 650 species name must have a lower case 

 Corrected

6. 753, 787 Do not capitalise the words in the titles of journal articles ( even if this was done in the original journal, as some do do) 

 Corrected

7. 766 there are 2 Smith refs here, the second one needs a new line and new number 

 Corrected- the second Smith reference was inserted accidentally.

8. 776, 788 you need to abbreviate this journal title for consistency

 Corrected

Reviewer #2:

Thank you for your thoughtful and thorough comments on this manuscript. We have addressed each of your requested changes below:

1. This paper reports a study examining changes in Hawaiian shoreline communities in relation to sea surface temperature. The paper potentially offers new and useful information on the ecology of these systems. However, in its current form, the paper is not reviewable. Although the core of the paper is the abundance of shoreline species, key figures with these data are missing in my copy, specifically Supplementary Tables 6 and 7 and Figure 6.

 Our sincerest apologies for the omission of these tables and figure. It was an error that occurred during the final formatting and submission, and the updated manuscript is uploaded to include the missing items. Additionally, for your convenience these figures/tables and their legends are included in the Response to Reviewer's letter, which is included at the end of the resubmitted Manuscript Draft PDF.

2. Beyond missing figures, most figures in the paper only provide statistical summary lines/curves rather than the biological data themselves. It is essential that the underlying data also be provided so that the reader can assess the fits to the data. This is especially true of Figure 4, which offers relationships with the species abundance data, but is also true for Figs. 3 and 5.

 We have updated figures 3-5 to include the biological and SST data behind the statistical summary lines/curves. Additionally, Figure 3 was updated to display the non-parametric LOESS curve fit, rather than the linear, to mirror the updated Figure 2. 

3. More information in Figure 2 is also needed. Specifically, a linear temporal trend is provided, but visually the temporal patterns appears to exhibit an oscillatory pattern across the duration of the data (beyond the annual cycles). Perhaps there is a signal of a relationship to some large-scale oceanographic pattern such as the PDO that can be extracted, or perhaps using a non-linear regression with an annual cosine function, an unspecified cosine function on the order of 20 yr (make the period a variable to be fitted) and a linear function would clarify the magnitude of a potential trend. The concern is that a longer-period oscillation can look like a long-term trend and the pattern in the data indicates a distinctly non-linear trajectory through time. At the least, a non-parametric curve fit (LOESS, DWLS or similar) would be beneficial.

 Thank you for this important feedback regarding our analyses and visualizations of SST trends. To investigate the non-linearity of SST over the study’s timeframe, we updated our SST analyses to include a LOESS non-parametric curve fit, as suggested. We then clarified the change in SST trends from 2004—2011 to 2012—2019. Additionally, we updated our manuscript to acknowledge the potential influence of larger oceanographic patterns, such as PDO, and argue the relatively limited timeframe of our study (16 years) makes it difficult to adequately address their possible impacts on SST trends in Hawaiʻi within and beyond the timeframe of this study.

Reviewer #3

This is a very timely and interesting study that does very well in incorporating citizen science. I found the methods to be rigorous and well explained, findings and interpretations to be justified, and the manuscript to be very well written. I normally have more comments and feedback to authors, but in this case there's not much more to say. I look forward to seeing this article published.

Thank you for your kind and generous comment. We are so glad you enjoyed our study.

---

## [Decision Letter · Decision Letter 1]

16 Mar 2023

PONE-D-22-26540R1Spatial Variation and Antecedent Sea Surface Temperature Conditions Influence Hawaiian Intertidal Community StructurePLOS ONE

Dear Dr. Ward,

Thank you for submitting your manuscript to PLOS ONE. After careful consideration, we feel that it has merit but does not fully meet PLOS ONE’s publication criteria as it currently stands. Therefore, we invite you to submit a revised version of the manuscript that addresses the points raised during the review process. In particular, please consider the suggestions on statistical robustness of the results and the ideas for additional analysis to more comprehensively assess the dataset, as provided by the Reviewer (see below for more details).

We look forward to receiving your revised manuscript.

Kind regards,

Caroline Ummenhofer

Academic Editor

PLOS ONE

Journal Requirements:

Reviewers' comments:

Reviewer's Responses to Questions

**Comments to the Author**

1. If the authors have adequately addressed your comments raised in a previous round of review and you feel that this manuscript is now acceptable for publication, you may indicate that here to bypass the “Comments to the Author” section, enter your conflict of interest statement in the “Confidential to Editor” section, and submit your "Accept" recommendation.

Reviewer #1: All comments have been addressed

Reviewer #2: (No Response)

2. Is the manuscript technically sound, and do the data support the conclusions?

Reviewer #1: Yes

Reviewer #2: Yes

3. Has the statistical analysis been performed appropriately and rigorously? 

Reviewer #1: Yes

Reviewer #2: Yes

4. Have the authors made all data underlying the findings in their manuscript fully available?

Reviewer #1: Yes

Reviewer #2: Yes

5. Is the manuscript presented in an intelligible fashion and written in standard English?

Reviewer #1: Yes

Reviewer #2: Yes

6. Review Comments to the Author

Reviewer #1: this was already a nice paper on the first round and now the minor comments on that version have been corrected it is even better

Nice job andan interesting read!

Reviewer #2: The authors have addressed some of my concerns regarding the figures and the inference of SST patterns. I would be happier if they had used a non-linear model to extract the annual and the possible longer term oscillation and then looked for a trend, but at least the non-parametric smoothing is available to get a sense of the pattern and this is noted in the text. I feel the inclusion of the data points on the graphs is especially valuable. From the perspective of the PLoS ONE review criteria, I feel the information provided and analyses are sufficient and the conclusions drawn are reasonable, but will raise a few more points for the authors to consider that might strengthen the paper further.

First, I do worry about the high likelihood of erroneously identifying a relationship that goes along with applying an approach that makes a massive number of different statistical tests of association to find the “best” temperature window to relate to species changes in time and space (statistical fishing expedition). Normally one would require a severe correction of alpha values to control study-wide error rates (as is done in genomics, for example)—or in the case of model selection used here, large differences in relative support. It seems to me virtually certain using this type of procedure that associations will be found, whether real or due to chance. This type of analysis is largely unavoidable for the authors to explore latent responses, and I think that this insight outweighs the downsides. More broadly this issue can be avoided if the analysis is considered exploratory, and the authors do a good job of not reading much into the patterns they report. But it wouldn’t hurt to discuss this issue a bit more and emphasize that the results are exploratory, ideally generating specific hypotheses/predictions to test more rigorously in the future.

Second, I continue to feel that there is missed opportunity in the paper to take advantage of the rich biological data set that makes this a noteworthy contribution. For example, why not carry out a PCA or other dimension reduction analysis on the Reclassified Taxa, tossing out the rarest taxa but retaining the others to get a broader sense of community change, and compare the dominant axes to SST? Other interesting relationships may be uncovered of broader functional groups (implied by the axis loading) that jointly change with temperature than can be revealed by just looking at the six most dominant taxa cross sites. I was also curious about the response of native vs. invasive taxa, as there is some concern that the latter are disrupting Hawaiian shorelines (Hypnea musciformis, for example). One of course has to be careful here to qualify that invasives may be increasing simply through the dispersal/invasion process itself, but I do wonder if environmental change is a plausible contributor to driving invasion, although the dip in temperatures may help distinguish these effects.

7. PLOS authors have the option to publish the peer review history of their article (what does this mean?). If published, this will include your full peer review and any attached files.

Reviewer #1: **Yes: **Charles Griffiths

Reviewer #2: No

---

## [Author Response · Author response to Decision Letter 1]

28 Apr 2023

Dear Reviewer, 

Thank you for your helpful comments. Below we have addressed your new points of consideration, and our manuscript has been updated to reflect our responses to your suggestions. Additionally, for your convenience any edited sections of text are included in the Response to Reviewer's letter, which is included at the end of the resubmitted Manuscript Draft PDF.

1. "First, I do worry about the high likelihood of erroneously identifying a relationship that goes along with applying an approach that makes a massive number of different statistical tests of association to find the “best” temperature window to relate to species changes in time and space (statistical fishing expedition). Normally one would require a severe correction of alpha values to control study-wide error rates (as is done in genomics, for example)—or in the case of model selection used here, large differences in relative support. It seems to me virtually certain using this type of procedure that associations will be found, whether real or due to chance. This type of analysis is largely unavoidable for the authors to explore latent responses, and I think that this insight outweighs the downsides. More broadly this issue can be avoided if the analysis is considered exploratory, and the authors do a good job of not reading much into the patterns they report. But it wouldn’t hurt to discuss this issue a bit more and emphasize that the results are exploratory, ideally generating specific hypotheses/predictions to test more rigorously in the future. "

 Thank you for addressing this aspect of our research design. Indeed, in traditional frequentist statistical hypothesis testing a typical approach to the “multiple test problem” is to adjust the alpha-critical value (e.g., Bonferroni Correction) by the number of comparisons being made. In a model selection approach, as we employed here, the goodness-of-fit among alternative models to the data is calculated utilizing the Akaike Information Criterion derived from the Maximum Likelihood estimate, and penalizes models which use more independent variables (parameters) as a way of over-fitting (e.g., see Burnam and Anderson, 2002). In our case, this over-fitting based on number of models parameters is not an issue, as we are only testing for model fit between the observed temperature data and the biological response (either abundance or diversity). However, we agree with the reviewer that the climwin approach of testing for the fit of multiple temperature time-windows on either the abundance or diversity response variable creates many models for comparison. While we understand why this all-subset model selection approach can be viewed as a “fishing expedition”, the alternative of a prior identification of a sub-set of candidate models (temperature time-windows) is also problematic (e.g., Burnham and Anderson 2004; Dochtemann and Jenkins, 2010; Burnham et al., 2010), especially when prior work is not available to generate a meaningful subset of models for the relationship between SST and abundance/diversity of these Hawaiian intertidal taxa. Further, while our analyses do have an exploratory nature, we believe the insights gained from this approach provide a stepping stone for future hypothesis testing on the possible effects of SST in Hawaiian intertidal communities. In our discussion, we have now more clearly stated this point. 

 References cited 

 Burnham KP, Anderson DR (2002) Model selection and multimodel inference, 2nd edn. Springer, New York 

 Burnham KP, Anderson DR (2004) Multimodel inference: understanding AIC and BIC in model selection. Sociol Methods Res 33:261–304 

 Burnham KP, Anderson DR, Huyvaert KP (2010) AICc model selection in the ecological and behavioral sciences: some background, observations and comparisons. Behav Ecol Sociobiol. doi:10.1007/s00265-010-1029-6

 Dochtermann NA, Jenkins SH (2010) Developing multiple hypotheses in behavioral ecology. Behav Ecol Sociobiol. doi:10.1007/ s00265-010-1039-4

2. "Second, I continue to feel that there is missed opportunity in the paper to take advantage of the rich biological data set that makes this a noteworthy contribution. For example, why not carry out a PCA or other dimension reduction analysis on the Reclassified Taxa, tossing out the rarest taxa but retaining the others to get a broader sense of community change, and compare the dominant axes to SST? Other interesting relationships may be uncovered of broader functional groups (implied by the axis loading) that jointly change with temperature than can be revealed by just looking at the six most dominant taxa across sites. I was also curious about the response of native vs. invasive taxa, as there is some concern that the latter are disrupting Hawaiian shorelines (Hypnea musciformis, for example). One of course has to be careful here to qualify that invasives may be increasing simply through the dispersal/invasion process itself, but I do wonder if environmental change is a plausible contributor to driving invasion, although the dip in temperatures may help distinguish these effects. "

 Thank you for your thoughtful considerations on the use of biological data and potential analyses for expanding this research. We have several points of response to your comment and will try to address each of them thoroughly below:

a. “For example, why not carry out a PCA or other dimension reduction analysis on the Reclassified Taxa, tossing out the rarest taxa but retaining the others to get a broader sense of community change, and compare the dominant axes to SST? Other interesting relationships may be uncovered of broader functional groups (implied by the axis loading) that jointly change with temperature than can be revealed by just looking at the six most dominant taxa across sites.”

Regarding community change and functional groups: It may be helpful to point out that a comprehensive analysis of community change with these biological data was conducted in a recent publication by several of our co-authors (La Valle, Schaefer, et al. 2020; cited as #24 in our manuscript). In this publication, the question of how these biological communities have changed spatially (sites, islands, exposure type, substrate type) and temporally (years and decades) was addressed. They found changes in community composition and abundance across years were found to be site-specific and list the taxa that significantly changed in abundance (reported in Table 6 in La Valle et al.). Based on these prior findings, we opted not to conduct ordination analyses in the present study, as the dominant loading axes are likely to be both hyper-variable across all twelve focal study sites and driven by the most abundant taxa, which is already a primary focus of the present study. However, we agree that future work on additional aspects of community-level change is warranted and we make these suggestions more explicit in the Discusssion (please see below).

b. “I was also curious about the response of native vs. invasive taxa, as there is some concern that the latter are disrupting Hawaiian shorelines (Hypnea musciformis, for example). One of course has to be careful here to qualify that invasives may be increasing simply through the dispersal/invasion process itself, but I do wonder if environmental change is a plausible contributor to driving invasion, although the dip in temperatures may help distinguish these effects.”

Regarding invasive taxa: Thank you for highlighting this important point regarding invasive species and their possible impact on Hawaiian shorelines. Similarly, we were curious about the impacts of invasive taxa and conducted exploratory analyses to address this explicitly in prior drafts of the manuscript. However, a few key considerations led us to decide not focus this study on the invasive species in this data set. We found the presence and abundance of invasive species (i.e., Hypnea musciformis, Gracilaria salicornia) across all intertidal sites to be highly variable, with low to zero abundance at most sites, and since the OPIHI data are most appropriate for monitoring the more abundant species (Cox et al. 2012) we did not feel confident in including the analyses we conducted focused on the invasive species in the final version of the manuscript. However, we do note, that there were two instances where invasive algae were present as a dominant taxon within an intertidal site. At Onekahakaha, Hawai’i Island, Gracilaria salicornia was consistently detected as a highly abundant taxon. At this location, our site-level analysis found the abundance of Gracilaria salicornia to increase with increasing SSTs (S6 Table). In contrast, while Hypnea musciformis was found to be a highly abundant taxon at Waipuʻilani, Maui, we found no significant relationship between its abundance and SST, and we now point out these results in the revised manuscript.

---

## [Decision Letter · Decision Letter 2]

10 May 2023

Spatial Variation and Antecedent Sea Surface Temperature Conditions Influence Hawaiian Intertidal Community Structure

PONE-D-22-26540R2

Dear Dr. Ward,

We’re pleased to inform you that your manuscript has been judged scientifically suitable for publication and will be formally accepted for publication once it meets all outstanding technical requirements.

Kind regards,

Caroline Ummenhofer

Academic Editor

PLOS ONE

Additional Editor Comments (optional):

Reviewers' comments:

Reviewer's Responses to Questions

**Comments to the Author**

1. If the authors have adequately addressed your comments raised in a previous round of review and you feel that this manuscript is now acceptable for publication, you may indicate that here to bypass the “Comments to the Author” section, enter your conflict of interest statement in the “Confidential to Editor” section, and submit your "Accept" recommendation.

Reviewer #2: All comments have been addressed

2. Is the manuscript technically sound, and do the data support the conclusions?

Reviewer #2: Yes

3. Has the statistical analysis been performed appropriately and rigorously? 

Reviewer #2: Yes

4. Have the authors made all data underlying the findings in their manuscript fully available?

Reviewer #2: Yes

5. Is the manuscript presented in an intelligible fashion and written in standard English?

Reviewer #2: Yes

6. Review Comments to the Author

Reviewer #2: (No Response)

7. PLOS authors have the option to publish the peer review history of their article (what does this mean?). If published, this will include your full peer review and any attached files.

Reviewer #2: No

---

## [Editor Report · Acceptance letter]

22 May 2023

PONE-D-22-26540R2 

Spatial Variation and Antecedent Sea Surface Temperature Conditions Influence Hawaiian Intertidal Community Structure 

Dear Dr. Ward:

I'm pleased to inform you that your manuscript has been deemed suitable for publication in PLOS ONE. Congratulations! Your manuscript is now with our production department. 

Kind regards, 

on behalf of

Dr. Caroline Ummenhofer 

Academic Editor

PLOS ONE